# Instruments to Measure Patient Satisfaction with Comprehensive Medication Management Services: A Scoping Review Protocol

**DOI:** 10.3390/pharmacy10060151

**Published:** 2022-11-14

**Authors:** Lorayne Caroline Resende, Mariana Martins Gonzaga do Nascimento, Mariana Michel Barbosa, Cristiane de Paula Rezende, Laís Lessa Neiva Pantuzza, Edna Afonso Reis

**Affiliations:** Center for Pharmaceutical Care Studies, College of Pharmacy, Federal University of Minas Gerais, Campus Pampulha, Belo Horizonte 31270-901, Minas Gerais, Brazil

**Keywords:** scoping review, medication therapy management, pharmaceutical services, patient satisfaction

## Abstract

Comprehensive medication management (CMM) is the service offered within the clinical practice of pharmaceutical care, which has the objective to optimize pharmacotherapeutic outcomes. Patient satisfaction is a multidimensional construct that points to the quality of the health services offered and the degree to which the patients’ expectations and needs are met. The evaluation of the level of patient satisfaction is a key indicator to support decisions and to improve the quality of the service provided. This study aims to describe the protocol for a scoping review to map the instruments to measure patient satisfaction with CMM services and compare them according to their development characteristics and the applicability of patient-reported outcome measures. The literature search will be conducted using the scoping review methodology, proposed by the Joanna Briggs Institute and the PRISMA Extension for Scoping Reviews (PRISMA-ScR) method. The results will be presented in two sessions: (1) description of the search strategy; and (2) the characteristics of the satisfaction instruments, number of items and questions related to the conceptual model, content validity, construct validity, reliability, score/interpretation, and respondent burden. This review will shed light on the available satisfaction measurement instruments, allowing existing gaps to be identified for future research.

## 1. Introduction

Pharmaceutical care represents a theoretical and methodological model for the clinical practice of pharmacists monitoring all pharmacotherapy in a systematic and organized way and having the patient as the center of care, aiming to optimize health outcomes. In this practice, the pharmacist works in individual consultations to identify and resolve drug therapy problems (DTPs). After the initial evaluation, the pharmacist develops care plans to solve or prevent the DTPs in collaboration with the patient and the health team, monitoring, in subsequent consultations, the results of each intervention and drawing up new plans whenever necessary [1].

Comprehensive medication management (CMM) is based in this theoretical and methodological framework of pharmaceutical care, which is a patient-centered practice that requires a close therapeutic relationship between the pharmacist and the patient [1,2,3,4]. Therefore, the CMM service quality and acceptability may be highly influenced by patient satisfaction [5]. The literature proposes the implementation of the quality and performance indicators to measure the quality of CMM services and propose improvements [2,5]. In a recent study, six key performance indicators for outpatient CMM services were developed and validated, among which the measurement of patient satisfaction was present [5].

The patient satisfaction measure is considered as a patient-reported outcome (PRO), which is a type of measure used to evaluate the results of patient-centered services and optimize decision-making about the services offered [6]. The concept of patient satisfaction does not a consensus in the literature, but it can be understood as a multidimensional construct to identify the quality of the health care service perceived by the user and the level to which their expectations and needs are met [7,8].

Among the main theories raised by Gill and White (2009) [9] to explain the concept of satisfaction are the discrepancy and transgression theory of Fox and Storms (1981), which points to satisfaction as the convergence of the patient’s health guidelines and the conditions of the provider [10]; Linder-Pelz’s (1982) expectation value theory, which considers satisfaction as a consequence of positive individual evaluations of different dimensions of health care [11]; the theory of the determinants and components of Ware et al. (1983), which addresses satisfaction mediated by the patient’s personal preferences and expectations [12]; the multiple model theory of Fitzpatrick and Hopkins (1983), which proposes that expectations are socially mediated and reflected in the patient’s health goals [13]; and Donabedian’s (1980) theory of health care quality [14].

The estimation of patient satisfaction is also recommended in the so-called “Quadruple Aim” as a guideline for institutions to seek to optimize the results of health services, which focuses on four dimensions: improving the health of the population; improving patient satisfaction with care; reducing health service costs; and improving the well-being at work of the health professionals who offer the service [15].

Despite the clear need to measure patient satisfaction for CMM services, the literature on the instruments available for CMM services is limited [16]. To verify if there have been any reviews on this topic, the authors of the present study carried out a previous search on the Medline and Cochrane databases in May 2021. This search did not retrieve any reviews on the topic. Therefore, to fulfill this knowledge gap, we intend to carry out a scoping review, as proposed by the Joanna Briggs Institute (JBI) [17], to map the existing literature on this topic. This methodology descriptively proposes examining the size, variety, and nature of the evidence when the subject is broad, complex, and too heterogeneous, as is the case of patient satisfaction with CMM [18].

In this context, this study aims to describe the protocol for a scoping review with the objective to map and analyze the instruments available to measure patient satisfaction with CMM services and compare them according to their development characteristics and the applicability of the patient-reported outcome measures. In other words, the scoping review that will be developed seeks to answer two questions: What satisfaction measurement instruments are available for patients treated for CMM services? and what are the characteristics of the instruments in relation to reliability, validity, and responsiveness?

In this context, the results of the scoping review may shed light on the available patient satisfaction instruments for measuring patient satisfaction, allowing existing gaps to be identified and analyzed for future research that will support the construction of new tools.

## 2. Materials and Methods

This protocol for the scoping review was designed following the principles set by the Preferred Reporting Items for Systematic Reviews and Meta-Analyses Extension for Scoping Reviews (PRISMA-ScR) [18] and the Joanna Briggs Institute (JBI) Manual for Evidence Synthesis [17]. The research protocol was properly registered on the Open Science Framework (OSF) platform on 25 February 2022, under number 10.17605/OSF.IO/RZA38 [19].

### 2.1. Pilot Search and Search Strategy Definition

Terms related to pharmaceutical services such as “medication therapy management” and “comprehensive medication management” and related to patient satisfaction such as “patient satisfaction” and “personal satisfaction” were included in the search. The presence of a satisfaction measurement instrument in the study will be evaluated in the inclusion criteria.

Initially, in November 2021, a pilot search was carried out using the MEDLINE and EMBASE databases to identify the keywords and indexing terms of interest. With the results of this pilot search, a screening by title and abstract of a random sample of approximately 5% of the studies retrieved was performed by two independent reviewers (L.C.R.; C.P.R.). The Rayyan^®^ platform was used in its blind setting to support study selection in this pilot [20]. The agreement between the two reviewers was assessed using the Cohen’s Kappa coefficient (k), which was calculated using the data retrieved from the Rayyan^®^ platform and transferred to Microsoft Excel^®^ [21]. The disagreement between the two reviewers in this pilot analysis was settled by a third independent reviewer (M.M.G.N.).

After the pilot, the search strategy was validated online in the Epistemonikos^®^ in December 2021 using the terms “medication therapy management” and “patient satisfaction”. The validation of terms using Epistemonikos^®^, which is a comprehensive systematic review database, has been recommended to optimize and increase the sensitivity of the search strategies, in order to ensure that no article of interest was not retrieved [22,23].

The same search terms were also validated online on the Google Scholar^®^ platform. The first ten pages of results on this platform were evaluated by the main researcher. According to recent studies, Google Scholar^®^ presents high overall coverage and acceptable precision, compared to MEDLINE [24]. Therefore, because of its incomplete coverage, this platform should not be used as a single source in systemic searching, but it may be a relevant additional source because it may retrieve studies non-indexed in the classical databases.

After the pilot search and validation stages performed on Epistemonikos^®^ and Google Scholar^®^, the main search terms were defined.

### 2.2. Study Search, Screening, and Selection

During the study’s search stage, the search terms previously defined will be combined in search strategies specifically designed according to the functionality and rules of each of these databases: MEDLINE, Latin American and Caribbean Health Sciences Literature (LILACS), EMBASE, Cochrane, Scopus, Cinahal, and Web of Science. An additional search in the gray literature will be performed in the Brazilian Catalog of Theses and Dissertations and in the Digital Library of Theses and Dissertations. Furthermore, to identify relevant studies that were not retrieved during the initial search, all references of the studies selected by the inclusion criteria will be reviewed.

First, the search results from all of the databases will be united in the Mendeley^®^ software, which will aid in the identification and exclusion of duplicates. After this, the evaluation of studies will be performed according to the eligibility criteria by the same pair of independent reviewers that performed the pilot search (L.C.R., C.P.R.). Both the reviewers will independently evaluate the content of the studies in the following order: (i) titles and abstracts; and (ii) full-text reading of the studies selected in the previous phase. A third reviewer (M.M.G.N.) will resolve disagreements by evaluating the full-text of the studies. The exclusion of studies in this first evaluation will be justified. The Rayyan^®^ platform will be used in its blind setting to support the screening and selection of the studies as well as the arbitrage performed by the third reviewer at this stage [20].

Following the PCC (Participants, Concept, and Context) strategy suggested for scoping reviews, the adopted inclusion criteria will be: (i) Participants—adult patients, or their caregivers, followed in a CMM service, which is based on a patient-centered care practice that involves comprehensive evaluation of each medication in use to determine its necessity, effectiveness, safety, and aligned with the definition of the Patient-Centered Primary Care Collaborative presented previously. Thus, the service must include an individualized plan to achieve pharmacotherapy goals with follow-up to determine the patient outcomes [2]; (ii) Concept—instruments to measure patient satisfaction with CMM services; and (iii) Context—all clinical practice settings in any geographic region. In addition, the following inclusion criteria will be adopted: (i) year of publication—studies published from the year 1990, considering this was the year in which the Pharmaceutical Care practice was proposed by Hepler and Strand [25]; and (ii) type of studies—qualitative, experimental, quasi-experimental, observational, descriptive, or methodological studies that describe the development of patient satisfaction instruments. There will be no language restrictions.

The following exclusion criteria will be adopted: (i) type of publication—book chapters, letters, editorials, notes, abstracts with insufficient information for eligibility analysis; (ii) study type—reviews, study protocols, study projects; (iii) type of participants—studies involving health professionals; (iv) type of pharmaceutical service—medication reconciliation, transition of care, disease state management, medication review, guidance on correct medication use, pharmaceutical education, health condition screening or dispensing services; (v) studies that did not address the development and/or use of a patient satisfaction instrument; (vi) studies not fully available; and (vii) satisfaction measure instruments not fully available. Instruments not made available even after request will be excluded from the evaluation.

### 2.3. Data Extraction and Synthesis of Results

Two independent reviewers will perform the data extraction using spreadsheets created by the authors in Microsoft Excel^®^ software. Data extraction spreadsheets and contents will be validated in consensus by two reviewers (L.C.R.; C.P.R.) using two studies that meet the inclusion criteria and validated by all other researchers involved in the review.

These groups of data will be extracted in duplicate: (i) characteristics of the studies (main author and year of publication; country; title; objective; context; type of pharmaceutical service; service definition; service reference in the literature; study design; number of participants); and (ii) characteristics of the patient satisfaction measurement instruments (name of the instrument; instrument’s objective; country and year of its development; application format–printed or electronic; response options—Likert scale, open field, dichotomous, mixed; administration method—self-applied, service provider, third party) number of instrument items; and questions related to the conceptual model, content validity, construct validity, reliability, score/interpretation, and respondent burden. Instruments not publicly available will be requested from their authors. Disagreements will be resolved after reaching consensus by both reviewers.

### 2.4. Critical Appraisal of Patient Satisfaction Instruments

The critical appraisal of the instruments identified will be carried out using a checklist developed by Francis et al. (2016) with the objective to systematically evaluate the PRO (patient-reported outcome) measures and to recognize their applicability, strengths, and weaknesses. The checklist contains 18 items covering seven domains: (i) conceptual model; (ii) content validity; (iii) reliability; (iv) construct validity; (v) scoring and interpretation; (vi) respondent burden; and (vii) presentation [26].

Each item will receive 0 or 1 points according to the absence or presence of the criterion in question in the instrument. The final score for each instrument will be the sum of the individual item scores.

### 2.5. Synthesis of Results

The results will be presented in two broad sections. The first section will describe the included studies through the PRISMA Flow Diagram for Scoping Reviews in the version published in 2020. The second section will present the most relevant findings from the scoping review in the form of diagrams and tables, allowing for the visualization of the characteristics of the identified satisfaction measurement instruments. Data from the quantitative assessment of the instruments will be shown in a table with the final score of each instrument according to the topics evaluated.

## 3. Preliminary Results

During the pilot search, a total of 2503 studies was retrieved (1531 in the MEDLINE database and 972 in the EMBASE). After the independent evaluation of 122 studies randomly selected, 95 studies were excluded and 22 were included by both reviewers, but they did not reach agreement for five studies. Therefore, the agreement between the two reviewers reached 95.9%, with a Cohen’s Kappa coefficient of 0.872, demonstrating very good agreement strength. After the evaluation of a third independent reviewer, 101 studies were excluded and 21 were included.

After the pilot, the validation search performed in the Epistemonikos^®^ database retrieved 46 studies. These studies were all related to the review’s questions. These same studies had also been retrieved during the pilot search.

In the Google Scholar^®^ platform, a total of 811,000 studies were retrieved. After the analysis of the first ten pages of results, only four relevant studies were identified. These studies were not retrieved by the pilot search strategy, but three of them were not published in scientific journals.

After this pilot and validation stage, the search terms previously defined were maintained in the final search strategy and another term was added: “clinical pharmacists”. All of the terms established for the scoping review from the “pharmaceutical services” and “patient satisfaction” groups are presented in Table 1.

## Figures and Tables

**Table 1 pharmacy-10-00151-t001:** Search terms defined for the scoping review.

Group 1—Pharmaceutical Services	
Medication Therapy Management (Mesh/Emtree) MTM services Management, Medication Therapy Therapy Management, Medication Drug Therapy Management Management, Drug Therapy Therapy Management, Drug Pharmaceutical Services (Mesh) Services, Pharmaceutic Services, Pharmacy Pharmaceutical Services Pharmaceutic Service Service, Pharmaceutic Services, Pharmaceutical Pharmaceutical Service Service, Pharmaceutical Pharmacy Services Pharmacy Service Service, Pharmacy Pharmaceutical Care (Emtree) Care, Pharmaceutical Comprehensive Medication Management CMM services Pharmacy Service, Clinical Service, Clinical Pharmacy Clinical Pharmacy Services Pharmacy Services, Clinical Services, Clinical Pharmacy Clinical Pharmacy Service Clinical Pharmacists (Emtree)	Community Pharmacy Services (Mesh) Pharmaceutical Service, Community Pharmaceutical Services, Community Service, Community Pharmaceutical Services, Community Pharmaceutical Pharmacy Services, Community Community Pharmacy Service Pharmacy Service, Community Services, Community Pharmaceutic Services, Community Pharmacy Community Pharmaceutical Services Community Pharmaceutic Service Pharmaceutic Service, Community Pharmaceutic Services, Community Service, Community Pharmaceutic Community Pharmaceutical Services Community Pharmaceutical Service Service, Community Pharmacy Medication Treatment Conduct Medication Management Drug Therapy Management Pharmacological Treatment Management Medication Treatment Management Community Pharmacy Services Medication Therapy Management pharmaceutical care Pharmaceutical Care Pharmaceutical attention Pharmaceutical Assistance Services Evidence Based Pharmaceutical Care Evidence-Based Pharmaceutical Practice Administration of Pharmacological Treatment Pharmaceutical Services Community Pharmacy Services Evidence-Based Pharmaceutical Practice
**Group 2—Patient Satisfaction**	
Patient Satisfaction (Mesh/Emtree) Satisfaction, Patient Personal Satisfaction (Mesh) Satisfaction, Personal patient satisfaction survey Validated patient satisfaction survey Patient satisfaction questionnaire (emtree) Satisfaction survey Patient experience (emtree)	Patient satisfaction Patient Satisfaction Patient satisfaction questionnaire Patient satisfaction survey Satisfaction survey Patient satisfaction survey Patient satisfaction questionnaire

## Data Availability

Not applicable.

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
