# Peer review of "Instruments to Measure Patient Satisfaction with Comprehensive Medication Management Services: A Scoping Review Protocol"

_pharmacy, 2022, doi:10.3390/pharmacy10060151_

Round 1

Reviewer 1 Report

it is recommended to describe in greater detail the results indicated in Summary 1) descripcion of the research strategy and 2) the characteristics of the satisfactions instruments and thus serve as a reference for future research.

I understand the complexity of the subject that is presented, so I want the authors to be able to answer the questions they raised, so that readers have a more accurate knowledge of the research work.

·        What satisfaction measurement instruments are available for patients treated for Comprehensive Medication management (CMM) services?

·        What are the characteristics of the instruments in relation to reliability, validity, and responsiveness?

In order to fulfill the stated purpose: This review will shed light on the available satisfactions measurements, allowing to identify and analyze existing gaps for future researches that will support the construction of new tools 

Reviewer 2 Report

I think this is a valid area of inquiry with increasing importance. That being said, I find that the literature referenced is quite old, and more recent literature isn't referenced. Here are some comments.

- you start your introduction talking about pharmaceutical care as if the primary audience for this manuscript isn't pharmacists when it is. This needs to be rewritten with the audience in mind. 

- on lines 55-57 you say "The literature proposes..." and then don't provide any references to literature. 

- on lines 62-65 you state that the concept of patient-satisfaction is not a consensus in the literature, but then you give a reference that's 10 years old. There is more recent literature in this area that should be consulted and referenced. 

- on lines 82-84 you say that a previous survey on Medline and Cochrane databases was carried out in May 2021, but by who? Where is it presented? 

- why did you only use patient satisfaction and personal satisfaction as search terms? What about consumer/customer/client satisfaction? 

- why were book chapters excluded? 

Round 2

Reviewer 2 Report

Thank you for your responses to my questions and suggestions. As someone who is not from the US I still find that the introduction is too basic for the audience but I will not insist it be changed. I also find that while I understand on a superficial level why you said you didn't use ' consumer/customer/client satisfaction' if it's combined with pharmacy or pharmacist or even health care you could eliminate the non-health related articles; however, I will not insist on this change if the editors are fine with it.